# The Antimicrobial Properties of Poplar and Aspen–Poplar Propolises and Their Active Components against Selected Microorganisms, including *Helicobacter pylori*

**DOI:** 10.3390/pathogens11020191

**Published:** 2022-01-31

**Authors:** Jarosław Widelski, Piotr Okińczyc, Emil Paluch, Tomasz Mroczek, Jakub Szperlik, Magdalena Żuk, Zbigniew Sroka, Zuriyadda Sakipova, Ioanna Chinou, Krystyna Skalicka-Woźniak, Anna Malm, Izabela Korona-Głowniak

**Affiliations:** 1Department of Pharmacognosy with the Medicinal Plant Garden, Medical University of Lublin, 20-093 Lublin, Poland; 2Department of Pharmacognosy and Herbal Medicines, Wrocław Medical University, 50-556 Wroclaw, Poland; zbigniew.sroka@umw.edu.pl; 3Department of Microbiology, Faculty of Medicine, Wrocław Medical University, 50-376 Wroclaw, Poland; emil.paluch@umed.wroc.pl; 4Department of Chemistry of Natural Products, Medical University of Lublin, 20-093 Lublin, Poland; tmroczek@pharmacognosy.org (T.M.); kskalicka@pharmacognosy.org (K.S.-W.); 5Faculty of Biological Sciences, Botanical Garden, Laboratory of Tissue Culture, University of Wrocław, 50-525 Wroclaw, Poland; jakub.szperlik@uwr.edu.pl; 6Faculty of Biotechnology, Wrocław University, 51-148 Wroclaw, Poland; magdalena.zuk@uwr.edu.pl; 7School of Pharmacy, S.D. Asfendiyarov Kazakh National Medical University, Almaty 050000, Kazakhstan; sakipova.z@kaznmu.kz; 8Division of Pharmacognosy and Chemistry of Natural Products, Department of Pharmacy, National and Kapodistrian University of Athens, 15771 Athens, Greece; ichinou@pharm.uoa.gr; 9Department of Pharmaceutical Microbiology, Medical University of Lublin, 20-093 Lublin, Poland; anna.malm@umlub.pl (A.M.); iza.glowniak@umlub.pl (I.K.-G.)

**Keywords:** propolis, LC-MS, principal component analysis, PCA, hierarchical fuzzy clustering analysis, MIC, MBC, galangin, pinocembrin, *Helicobacter pylori*, *Staphylococcus aureus*, *Staphylococcus epidermidis*, *Enterococcus faecalis*, *Bacillus cereus*, *Micrococcus luteus*

## Abstract

There is a noticeable interest in alternative therapies where the outcome is the eradication of the Gram-negative bacterium, *Helicobacter pylori* (*H. pylori*), for the purpose of treating many stomach diseases (chronic gastritis and peptic ulcers) and preventing stomach cancer. It is especially urgent because the mentioned pathogen infects over 50% of the world’s population. Recent studies have shown the potential of natural products, such as medicinal plant and bee products, on the inhibition of *H. pylori* growth. Propolis is such a bee product, with known antimicrobial activities. The main scope of the study is the determination of the antimicrobial activity of ethanolic extracts from 11 propolis samples (mostly from Poland, Ukraine, Kazakhstan, and Greece) against *H. pylori,* as well as selected bacterial and yeast species. The most effective against *H. pylori* was the propolis from Ukraine, with an MIC = 0.02 mg/mL while the rest of samples (except one) had an MIC = 0.03 mg/mL. Moreover, significant antimicrobial activity against Gram+ bacteria (with an MIC of 0.02–2.50 mg/mL) and three yeasts (with an MIC of 0.04–0.63 mg/mL) was also observed. A phytochemical analysis (polyphenolic profile) of the propolis samples, by ultra-high-performance liquid chromatography-diode array detector-mass spectrometry (UPLC-DAD-MS), was performed. An evaluation of the impact of the propolis components on antimicrobial activity, consisting of statistical analyses (principal component analysis (PCA) and hierarchical fuzzy clustering), was then performed. It was observed that the chemical composition characteristics of the poplar propolis correlated with higher antibacterial activity, while that of the poplar and aspen propolis correlated with weaker antibacterial activity. To summarize the activity in vitro, all tested propolis samples indicate that they can be regarded as useful and potent factors in antimicrobial therapies, especially against *H. pylori.*

## 1. Introduction

*Helicobacter pylori* (*H. pylori*) is a Gram-negative, spiral-shaped, microaerophilic bacterium usually found in the digestive tract, especially in the stomach [1,2]. Moreover, *H*. *pylori* colonize the stomachs of more than 50% of the worldwide population and have been implicated in the pathogenesis of several digestive tract disorders, such as chronic active gastritis, peptic ulceration, gastric cancer, and mucosa-associated lymphoid tissue (MALT) lymphoma [3]. The development of a spectrum of ailments related to the presence of the aforementioned pathogen depends on the strain’s virulence, the efficiency of the host’s immune system, and many factors connected to the environmental conditions, e.g., the sanitation standards, diet, or addictions [4,5]. The conventional eradication therapy for *H. pylori* infections includes a proton pump inhibitor (PPI) containing the antibiotics clarithromycin and amoxicillin together with metronidazole (triple therapy), which remains the first-choice therapy [6]. This way of treatment has several disadvantages. Chemotherapeutics used in the eradication of *H. pylori* present numerous adverse effects, among them nausea, vomiting, headaches, general malaise, dizziness, and the long-term perturbation of intestinal microbiota [7]. Another inherent problem of antibiotic-based therapies is the increasing resistance of *H. pylori* towards commonly used antibiotics, especially clarithromycin, levofloxacin, and metronidazole [8]. Worldwide growing antimicrobial resistance of the described pathogen is a huge medical problem for physicians, as well as society. It is reflected by the indecision of the WHO, which ranked *H. pylori* as priority 2 on its global list of antibiotic-resistant bacteria [6,9]. Therefore, there is an urgent need to find new drugs with potential activities against *H. pylori*, which will allow for an effective, safe, and preferably inexpensive therapy. Several therapeutic alternatives beyond antibiotics, focusing on vaccines, phage therapy, and probiotics, have been tried out in the last years [10]. There are high expectations for the use of phytotherapy in the treatment of *H. pylori* infections. Natural products with low toxicity, improved stability, easy availability, and a relatively low cost seem to be excellent candidates for developing future anti-*H. pylori* therapeutics, as well as leading compounds [2]. Moreover, traditional use and both in vitro and in vivo studies have confirmed the successful application of many natural products, among them mastic gum, broccoli, blueberries, cinnamon, curcumin [6], and thyme, as well as lemongrass, cedarwood, and lemon balm essential oils [10] and *Chelidonium maius* [5] in *H. pylori* eradication therapy. It seems that bee products, mainly propolis and honey, are an excellent alternative therapy for *H. pylori* eradication. Propolis, also called bee glue, is a natural product, a resinous substance produced by honeybees from buds and their different exudates. Its main role is to protect and maintain the beehives’ structural integrity (construction material), as well as providing for an aseptic internal environment of the hive (antimicrobial agent) [11]. In general, propolis is a mixture consisting of resin (50%), wax (30%), essential oils (10%), and 5% of other organic compounds [12]. Among the all-natural substances, propolis is a particularly rich source of phytochemicals of the polyphenolic group, mostly phenolic acids, flavonoids, and their derivatives [6]. Different caffeic acid esters, such as *p*-coumaric acid, chrysin, galangin, pinocembrin, pinostrobin, and quercitin are the frequent and usually the most abundant components of propolis [13,14]. Fortunately, these compounds are the main bioactive molecules responsible for the biological activity of propolis and are indicators of the origin of the propolis sample, concerning the geography and the plant source [6,11,13].

The main goals of this study were to:

Evaluate the anti-*H. pylori* activity of tested propolis samples of different origins;Assess the relation between the polyphenolic profile of propolis, their plant origins, and their antimicrobial activity;Evaluate the general antimicrobial profile of the tested propolis as a way to better understand the activity of the tested propolis extracts against *H. pylori.* For this purpose, the antibacterial activities of propolis extracts against six Gram+ and five Gram− (not including *H. pylori*) bacterial strains, as well as three species of yeasts, were evaluated.

## 2. Results and Discussion

### 2.1. The Chemical Composition and the Classification of Tested Propolis Samples

The results of the chemical analysis of the extracts obtained from the tested propolis samples are presented in Table 1. Among the main components of the investigated propolis, *p*-coumaric acids and its propyl ester, chrysin, pinocembrin, acacetin, pinocembrin chalcone, lasiocarpin A (1-acetyl-1,3-di-*p*-coumaroyl glycerol), galangin, 3-*O*-acetyl-pinobanksin, and pinostrobin were observed. In general, eighty-six compounds, as constituents of propolis, were identified or tentatively identified. The identification was based on comparison with standards, such as the MS and UV spectra of the substances, which is a well-established method that is described in the literature. Notably, most of the components were isolated previously from different types of propolis, or its plant precursors, in the frame of phytochemical research.

Generally, five main chemical groups were observed: flavonoids, free cinnamic acids, cinnamic acids monoesters, phenolic acids glycerides, and other polyphenolic compounds. The results of the analysis of the tested propolis samples were published in previous papers [13,15]. Only the propolis obtained from Parga (Greece) was analyzed by UPLC-DAD-MS for the first time. Chrysin, caffeic acid phenyl ester, galangin, and pinobanksin-3O-acetate were determined to be the main components.

### 2.2. The Antimicrobial Activity of Tested Propolis Samples

#### 2.2.1. The Antibacterial Activity of Tested Propolis Extracts against *H. pylori*

Propolis, as a natural substance with antimicrobial activity, has been traditionally used in folk medicine. Presently, it has been proven that bee glue possesses antiseptic and antimicrobial properties [16,17]. It can be extensively used in food, beverages, and food supplements for improving health and the prevention of many diseases, and it is also the source of new substances for future therapies [18].

We screened 10 different propolis extracts (PE) obtained from propolis samples from different geographical origins. Four samples came from Poland, four from Ukraine, one from Greece, and one from Kazakhstan. The highest activity against the referential *H. pylori* strain was shown by the PE from Ukraine (UK3) with a MIC an value of 0.02 mg/mL. Moreover, all tested PEs possessed good bioactivity, as their MICs were in the range of 26 to 125 µg/mL, according to the bioactivity criteria established by O’Donnell and colleagues [19]. For most of the PEs evaluated against *H. pylori*, MIC = 0.03 mg/mL was equal to MBC, giving an MBC/MIC ratio of 1, which confirmed the bactericidal activity of the tested PEs (see Table 2). Only the PLS2 sample exerted a higher MIC and MBC value compared to the majority of Pes, which amounted to 0.06 mg/mL (exactly 62.5 µg/L). Nevertheless, this still represents promising bioactivity.

The activities of the propolis extracts, isolated compounds, and volatile fractions against *H. pylori* were reported previously in several papers concerning samples of different origins and chemical compositions. Bonvehí et al. [20] evaluated the activity of 19 propolises collected from different locations throughout Basque Country (Spain) showing very weak activity against *H. pylori* (MIC from 6 to 14 mg/mL). In another study, 70% ethanolic extract was obtained from raw propolis produced by *Trigona* spp. bees from Indonesia and this was tested on 10 types of *H. pylori* clarithromycin- and metronidazole-resistant strains isolated from dyspeptic patients [12]. The MIC values of the PE were in the range of 1024 to 8192 µg/mL. According to the criteria the authors used for the evaluation of the antimicrobial activity, the tested PEs showed quite high antibacterial activity against 60% of the *H. pylori*-resistant strains, while the remaining strains have been shown only to be weakly inhibited by propolis [12]. These criteria for the antimicrobial activity of plant extracts were proposed by Tamakou et al. [21]. Very high cut-off points for the antimicrobial activity were created for edible or medicinal plants, which are treated equally with toxic plant extracts by the majority of scientists. Nevertheless, most of the systems would justify the application of samples with the highest MIC values (10–20 mg/mL) as non-active or as having very weak activity [19,22]. 

Very strong anti-*H. pylori* activity was exhibited by the polar extract of Nigerian propolis [23]. The PE was tested in vitro against clinical and reference (ATCC 43504) *H. pylori* strains. Moreover, the PE exhibited a similar inhibitory activity on both strains, with an MIC = 25.1 µg/mL and an MBC = 95.3 µg/mL, which suggests bactericidal activity. Additionally, the MIC value for propolis was eightfold higher than that for the antibiotic amoxicillin (the positive control) [23]. From the Nigerian propolis, several polyphenols were isolated: vesticarpan, medicarpin, vestigial, and 8-prenylnaryngenin, and their structures were confirmed by NMR and MS data [23]. The results of the aforementioned research confirm studies linking the presence of polyphenols in PEs with antibacterial activities against *H. pylori* [24].

Testing the antimicrobial activity of propolis samples are usually focused on the MIC and MBC values, but unfortunately, comprehensive or even general chemical analyses are missing. Research conducted by Romero et al. [6] is, in contrast to this general background, an exceptional work. The authors detected and identified twenty-one polyphenols in the tested Chilean propolis samples with chrysin, galangin, pinocembrin, and caffeic acid phenylethyl ester as the main target compounds. Moreover, a chromatographic quantitative analysis using HPLC-DAD established caffeic acid phenylethyl ester as an ingredient with the highest concentration in the samples, followed by pinocembrin and chrysin [6]. The four main polyphenol compounds of the tested propolis samples were isolated by a countercurrent chromatography technique (CPC) combined with preparative HPLC. The main components of the Chilean propolis are also present in the samples analyzed in this publication, which explains the relatively strong (UK3 sample), or the very good activity, against *H. pylori* found in the investigated samples. Surprisingly, the main compounds isolated from the propolis sample from Chile showed moderate to mild anti-bacterial activity upon the tested *H. pylori* strains [6]. The exception was a mixture of chrysin and galanin (FIC_index_ = 0.14) [6]. The above results suggest that the activity of propolis is associated with the synergy of all polyphenols and that propolis with good anti-*H. pylori* activity is characterized by the presence of high amounts of chrysin and pinocembrin. This corresponds with our own results, which shows the high content of these flavonoids in all analyzed samples, resulting in their significant bioactivity. According to the research of Romero et al [6] that was confirmed by a morphological study through transmission electron microscopy, the diverse effect of polyphenolic propolis compounds on the bacterial cell ultra-structure was shown, including lysis, membrane vesicle formation, and membrane alteration, which may explain the mechanism of propolis activity against *H. pylori*.

Antioxidant properties of the tested propolis samples described in our previous work may have crucial importance for the eradication of *H. pylori* [13,25]. Infections caused by *H. pylori* are preceded by gastric mucosa colonization, which is connected with the production of reactive oxygen and nitrogen forms. Therefore, natural products, such as propolis, which have the ability to reduce the production of harmful free radicals, can be regarded as a complementary therapy against pathogens [26]. This is especially so regarding the infection of *H. pylori*, which is associated with a low level of antioxidants in gastric juice. 

It seems that propolis, according to its multidimensional activities, can be a natural alternative for existing therapies that are becoming less effective due to bacterial resistance for patients with severe symptoms of *H. pylori* infections.

#### 2.2.2. The Antimicrobial Activity of Tested Propolis Extracts

The eleven tested PEs were evaluated for their antimicrobial activity against six Gram-positive bacterial strains, as well as three human pathogenic fungi. The results presented in Table 2 showed the interesting and promising antimicrobial activities of the tested PEs. Particularly promising were the results for the anti-staphylococcal activity against the two reference strains, *Staphylococcus aureus* ATCC 25923 and ATCC 29213 (MIC values 0.01–2.5 mg/mL). Very strong activity, according to O’Donnell criterium, was shown by one PE sample from Ukraine (UKT), with an MIC = 0.01 mg/mL (exactly 9.8 µg/mL) for both strains of *S. aureus*, an MBC = 0.04 mg/mL (exactly 39 µg/mL), and MBC/MIC ratio of 3.97, which indicates bactericidal activity. The Greek sample of propolis, obtained from Parga region, exhibited strong bioactivity with an MIC = 0.02 mg/mL and an MBC = 0.08 mg/mL for *S. aureus* ATCC 25923, as well as an MIC = 0.08 mg/mL and an MBC = 0.16 mg/mL for *S. aureus* ATCC 29213. Among the rest of the tested PEs, the propolis samples from Poland possessed good antibacterial bioactivity against *S. aureus*: PLS3, as well as Ukraine: UK2 and UK3. These excellent results that show the potent activity are corroborated by research concerning the antibacterial potential of Eastern European propolis. Grecka et al. [27] used ethanol PEs from 20 apiaries located in different regions of Poland for the determination of anti-staphylococcal activities. The MIC values were in the range of 256 to 512 µg/mL, which indicates a moderate activity against the tested *S.aureus* strains. Only one sample was shown to have a good bioactivity of 128 µg/mL, which was analogous to propolis samples from the Lublin region (MIC = 0.9 mg/mL) [11].

Several independent studies have confirmed the high susceptibility of *S. aureus* and *S. epidermidis* to Brazilian propolis. For example, Reguiera et al. confirmed the good activity of Brazilian red propolis against *S. aureus*, with an MIC in the range of the concentration of 64 to 1024 µg/mL [28]. Another group of scientists worked on the antimicrobial efficacy of three different types of propolis from Brazil: red, green, and brown, where the ethanolic extracts of the red type of propolis appeared to be the most active. Its MIC extended from 25 to 100 µg/mL, which indicates good antibacterial activity, and was similar to our results [29]. Moreover, the authors indicated low-pressure extraction (the classical type of extraction) with the use of ethanol as a better method of extraction, in comparison to supercritical fluid extraction (SFE) using CO_2_ as the supercritical fluid [29]. This is due to the fact that SPE is an extraction type that is suitable for less polar compounds compared to those that determine the activity of propolis, including its antimicrobial properties. These conclusions confirmed the validity of the choice of the mix of solvents (ethanol in water 70:30; *v*/*v*) and the extraction method (ultrasonic bath) used in the above study, as the appropriate method for obtaining polyphenolic compounds from investigated propolis samples. Impressive results relating to the activity of 39 propolis samples obtained from South Africa were published by Suleman et al. [30]. The MIC for the three most active propolis extracts was 6 µg/mL, and two of them had an MBC at the same concentration. The third was slightly higher at 9 µg/mL [30]. With such strong antibacterial bioactivity, African samples were only slightly more active than the Ukrainian samples from Tarnopol analyzed in this paper. Concerning the antimicrobial activity levels, most of the propolis from Europe, South America, and Asia displayed lower activities compared to the samples tested by our team. For example, the European propolis samples collected from various geographic origins (the Czech Republic, Ireland, and Germany) that were evaluated by Al-Ani et al. showed moderate bioactivity (an MIC from 0.08 mg/mL to 2.5 g/mL [31].

Tested PEs have shown very interesting activities against the *Staphylococcus epidermidis* ATCC 12228 strain (*S. epidermidis*). Propolis samples obtained from Tarnopol, Ukraine (UKT) exhibited very strong antimicrobial activity (MIC = 0.01 mg/mL) while the Greek propolis from Parga (GP) had strong activity (MIC = 0.02 mg/mL). Five samples of tested propolis had good bioactivity. Two of them, PLS3 and UK1, had MICs = 0.04 mg/mL, while two had slightly higher MIC values of 0.08 mg/mL (KZ, PLS 1, and UK3). The MIC values of the rest of the tested propolises were in the range of 0.16 mg/mL to 0.31 mg/mL, which provides evidence for the moderate antibacterial activity against *S. epidermidis*. Our results correspond to the MIC outcomes published in another paper concerning the antibacterial activity of PE that originated in Europe [11,27], but they are better. The MIC and MBC values of the eleven tested propolis samples were determined for the opportunistic bacteria, *Enterococcus faecalis* ATCC 2912 (*E. faecalis*). PEs obtained from Greek (GP) and Ukrainian (UKT) samples had good antibacterial activities (MIC = 0.08 mg/mL) and four propolis samples, PLS3, UK1, UK2, and UK3 had moderate activity (MIC = 0.16 g/mL). The rest of the samples were characterized by higher MIC values, showing their weaker activity. The obtained results are similar to that exerted by South African propolis (an MIC value of 49 to 1563 µg/mL) [30] and our previous research concerning *Apis mellifera* L. and *Trigona* sp. propolis from Nepal [32]. They are much better than those evaluated in 53 Serbian samples (an MIC range of 0.4 to 16.8 g/mL) [33]. The antimicrobial activity of PEs against *Micrococcus luteus* ATCC 1040 (*M*. *luteus*) is shown only in a handful of papers. Therefore, data is provided by our experiment concerning the above-mentioned activity. The Greek propolis (GP) had the highest activity against *M. luteus* (MIC = 0.02); the samples from Kazachstan (KZ) were less active than the Poland PLS3 and Ukraine (UK1-3) samples, with an MIC = 0.08 µg/mL. Four Anatolian propolis samples possessed better antimicrobial activity against *M. luteus* [34], with MIC values of 4 to 16 µg/mL. Those originating from Brazil were definitely less active than our samples. For the 12 fresh and aged propolis samples, the MIC values were between 340 and 650 µg/mL [35].

Tested propolis samples had a weaker activity against the well-known foodborne pathogen, *Bacillus cereus* ATCC 10876 (*B. cereus*). The best MIC value, 0.02 µg/mL, against this Gram+ bacteria was observed in the propolis from Parga, Greece (GP), which was slightly better than the activity of Korean propolis [36].

MIC and MFC values against three pathogen yeasts, *Candida albicans* ATCC 102231, *C. parapsiliosis* ATCC 22019, and *C. glabrata* ATCC 90030, for all tested propolises, were determined. It is worth mentioning the antifungal activity of propolis from Tarnopol (UKT) with an MIC = 0.02 mg/mL for *C. parapsiliosis*, and 0.08 mg/mL for the two other yeasts. The samples from Parga (GP), Poland (PLS3), and Ukraine (UK1–3) showed good antifungal bioactivities, with MICs between 0.04 to 0.16 mg/mL (see Table 2). Our results are similar to the activity of the Anatolian propolis for *C. albicans* (an MIC range of 4 to 32 µg/mL) [34] and is better than that of the propolis samples from South Africa (with an MIC between 98 and 1563 µg/mL] [30].

The differences in the biological activity of the propolis extracts depend on the collection region and flora specific to the region, as well as the species of the honeybees, the season of collection, and even the weather in a particular year. [11]. The underlying mechanism is linked with its chemical composition, especially with polyphenolic compounds. The strong bacteriostatic and bactericidal effects of propolis rely on its multidimensional and combined effects achieved by decreasing protein synthesis and preventing cell division. This inhibits the growth of microorganisms [37,38]. This activity is correlated with the high content of flavonoids, such as galangin, pinocembrin, and pinobanksin, as well as the esters of phenolic acids, which are well-known for their high antimicrobial activity [11,17].

### 2.3. Impact of Components on Antimicrobial Activity

Results are presented in Appendix A. Model 1—components vs. MIC, the impact of components on antibacterial activity; and Appendix A. Model 2—components vs. components). Statistical analyses in Model 1 have shown more significant (*p* > 0.05) observations of positive correlations (64) than negative correlations (15). However, the same negative impact on antibacterial activity was expressed as the positive value of the R factor (a positive correlation refers to increasing the parameter MIC accompanies, raising the component peak value) while the positive impact was described as the negative value of the R factor (a negative correlation refers to decreasing the parameters of the MIC with an increase in the component peak value). Model 1 included Gram-positive bacteria and fungi due to the too-low differences between MICs against the Gram-negative ones. 

Vanillin, 2-acetyl-1,3-di-*p*-coumaroylglycerol, and 2-acetyl-3-*p*-coumaroyl-1-feruloylglycerol had a negative impact on the antibacterial activity against all tested microorganisms. Multiple components, such as *p*-coumaric, ferulic, cinnamic, and benzoic acids also exhibited a negative impact on antibacterial activity against most of the tested microorganisms, while 2-acetyl-3-caffeoyl-1-feruloylglycerol only did so against *C*. *albicans*. 

Only some flavonoid aglycones were found to have a positive impact on activity, such as pinobanksin-5-methyl ether, quercetin-3-methyl ether, pinobanksin, chrysin, galangin, and pinobanksin-3O-acetate. However, this impact was observed only against some microorganisms, which included: *S. epidermidis* (chrysin), *E. faecalis* (all components), *B. cereus* (pinobanksin, quercetin-3-methyl ether, galangin, and chrysin), *C. parapsilosis* (pinobanksin-3O-acetate), and *C. glabrata* (all components).

Statistical analyses in Model 2 established that the presence (high peaks in the chromatogram) of one component with a negative impact on antimicrobial activity usually positively (*p* < 0.05 and R > 0) correlated with the presence of another component with a negative impact, and negatively correlated with components with a positive impact on antibacterial activity. For example, the presence of vanillin was positively correlated with the ferulic, benzoic, and cinnamic acids, as well as 2-acetyl-1,3-di-*p*-coumaroylglycerol and 2-acetyl-3-*p*-coumaroyl-1-feruloylglycerol. A negative correlation (*p* < 0.05 and R < 0) was found for pinobanksin-5-methyl ether, pinobanksin, chrysin, galangin, and pinobanksin-3O-acetate. The same situation was observed for components with a positive impact on antimicrobial activity.

It is important that the positive impact was observed only for the flavonoid aglycones, while the negative correlations were noted for the hydroxycinnamic acids, their glycerides, benzoic acids, and vanillin. Those results imply that the weaker antimicrobial activity of the propolis extracts is usually connected with the presence of a specific group of components. Therefore, some groups of components that are responsible for different types of propolis antimicrobial activities, in 70% ethanol with water extracts, may be connected to their plant origin.

We have found the abovementioned hypothesis to be worth testing. In the literature, we have found that investigations performed by different researchers established that the correlations between the composition and the activity is complex [39,40,41]. Usually, despite differences in chemical compositions, the same profile of propolises sourced worldwide possess very similar antimicrobial activities [39,40,41]. Differences were usually noted in the level of activity against various microorganisms [39,40,41,42]. However, some research showed that the general tendencies for some types of propolis can be described. Researchers tried to correlate the antimicrobial effects with the total amounts of polyphenols [39,42], groups of substances [40,42], and singular components [40,42]. Generally, most research setups included a poplar type of propolis and, usually, a higher correlation was exhibited with Gram-positive microorganisms than Gram-negative ones, as well as fungi [40,42]. This may be associated with the generally stronger sensitivity of the Gram-positive bacteria to xenobiotics [39].

The dependences between the composition and the antimicrobial activity were found to be variable. In the case of the total amount of polyphenols, a correlation was present [42] or absent [27,39,42] against different microorganisms [27,40,42]. 

Analyses of the groups of components were performed. Some researchers found a weak positive impact of the sum of free phenolic acids [40], the total amount of flavonoids [40], and the amounts of flavones, flavonols [42], flavanones, and dihydroflavonols [42]. 

These correlations were also present for singular components [40,42]. For example, the caffeic and ferulic acids exhibited a weak positive effect [40,42], or did not have any impact [42], on antibacterial activity. Stronger positive impacts on antibacterial activities were observed for some flavonoid aglycones (galangin, chrysin, pinocembrin, and pinobanksin-3O-acetate) [40] and phenolic acid monoesters (caffeic acid phenethyl ester and 1,1-dimethylallylester, usually known as caffeic acid prenyl ester) [40]. In our research, we observed positive impacts only for some of the singular flavonoid aglycones, probably because we did not observe a significant presence of phenolic acid monoesters in any of our samples. 

It is worth noting, however, that components that exhibited a negative impact on the antibacterial activity in our research, especially vanillin and benzoic acids, are known antimicrobial agents. Potentially, both substances performed worse than the flavonoids in the analysed compositions due to unknown interactions. Another explanation is the higher presence of these components next to weaker antibacterial agents, such as free phenolic acids and, probably, phenolic acid glycerides. Further research is required to acquire a better understanding of the observed phenomenon. 

### 2.4. Principal Component Analysis and Hierarchical Fuzzy Clustering

Results are presented in Figure 1 (principal component analysis) and Figure 2 (hierarchical fuzzy clustering dendrogram, two-factor projection). The principal component analysis demonstrated that the two-factor model explained 65.46% of the variability in the samples. A result of this strength is usually accepted as sufficient. Generally, no one component exhibited a dominant impact on factor compositions (Figure 1B). The projection of cases (samples) on the 2-factor plane (Figure 1A) obtained two main clusters and two subclusters in both the main clusters. This is important, as the first main cluster contained only samples with stronger antibacterial activities, while the second main cluster was composed of propolis samples with lower antibacterial activities. Moreover, it was impossible to obtain clusters with samples showing both stronger and weaker antimicrobial activities in the presented two-factor model (the geometrical angle between them was >90°, see Figure 1). Similar results were obtained in hierarchical fuzzy clustering analysis. Samples showing stronger and weaker antibacterial activities were divided into two different clusters. 

As shown by the analysis of the chemical composition, propolis with stronger antibacterial activity was composed of substances characteristic for poplar propolis, while propolis with weaker antibacterial activity was composed of components characteristic for poplar and aspen. It suggests that the addition of aspen mixes with the black poplar resins, decreasing the antibacterial effect of propolis. However, this requires further investigation.

The researched propolises belonged to the poplar and aspen–poplar types. Component characteristic for poplars included mainly flavonoid aglycones (galangin, chrysin, pinocembrin, and pinobanksin-3O-acetate) while the aspen markers included phenolic acid glycerides (mainly 2-acetyl-1,3-di-*p*-coumaroylglycerol). This division is known and is well-established in the literature [43,44,45,46,47]. We successfully used the principal component analysis (PCA) [15,25], as well as the hierarchical fuzzy clustering analysis (dendrograms) [9] for the classification of the plant origin of the 70% ethanol-in-water propolis in the previous publications. In the literature, different techniques were used to obtain data for propolis plant origin classifications, especially HPLC-MS [45,46], GC-MS after silylation [43,44], TLC-MS [47], and FT-IR [48]. The same classifications were usually obtained by PCA [15,25,45,46] or hierarchical fuzzy clustering (dendrograms) [43,44]. Our decision to use UPLC-DAD (UV), and not other analytical techniques, was forced by the nature of the propolis components. These are, mainly, polyphenols with good UV absorbance, yet not all of them produce a sufficient amount of ions in either negative or positive mass spectrometry modes. As a result, most poplar and aspen propolis components are best analysed by UPLC MS and are shown as UV chromatograms. Moreover, the HPLC technique allows for observation of the crude composition of propolis and does not produce a lot of artefacts.

Thanks to the comparison of antimicrobial properties and the plant origins, we have established that 70% ethanol-in-water extracts of poplar propolis were better antibacterial agents than the extracts from aspen–poplar propolises. A similar result was exhibited previously by Petri dish diffusion experiments [25]. Generally, both observations proved that the addition of aspen decreased the antimicrobial activity of ethanolic water extracts of propolis. However, Isidorov et al. [43] established that diethyl ether extracts of the propolis of birch, poplar, and aspen plant origins had a similar level of antibacterial activity. Moreover, the results of this research also proved that antimicrobial activity is not the only criterion for bees while collecting plant exudates. It is worth adding that, sometimes, the aspen propolis’ 70% ethanol-in-water extract [25], as well as the diethyl acetate extracts [43], are better antibacterial agents than some poplar ones. Probably, the main factor responsible for this phenomenon is the presence of several chemotypes of poplar trees [15,25,44]. A further implication of this fact may be the different impacts of this same component on antimicrobial activities in different propolis subtypes, especially of mixed plant origins.

## 3. Materials and Methods

### 3.1. Sample Preparation

Propolis was obtained from the following states: four samples from Poland (three from Lower Silesia and one from the Lubelszczyzna region), four samples from Ukraine (three from Khmelnitsky Village and one from Tarnopol), and one sample per state was collected from Kazakhstan (Almastka Oblast) and Greece (Parga).

The obtained propolis was frozen in liquid nitrogen and crushed in a mortar. Freezing and crushing were performed in triplicate (until the powdering of the propolis). The previously ground research material was extracted by ethanol in water (70:30; *v*/*v*) in the proportion of 1.0 g of propolis per 10 mL of solution. The extraction was performed in an ultrasonic bath (Sonorex, Bandelin, Germany). Extraction conditions were set at 40 °C for 45 min at 756 W (90% of ultrasound bath power). Extracts were then stored at room temperature for 12 h and, finally, were filtered through Whattman No. 10 filtrate paper (Cytiva, Marlborough, MA, USA). Zero point five milliliters of filtrated extract was used for the UPLC-DAD-MS analysis and the rest was evaporated and then lyophilized to dryness.

### 3.2. UPLC-DAD-MS Analysis of Propolis Extracts

The chemical compositions of the propolis extracts were analyzed by the Waters Acquity UPLC system (Waters, Milford, CT, USA) equipped with a PDA 200–500 nm, a mass spectrometer, Xevo-Q-TOF (Waters, Milford, CT, USA), and column BEH C18 130 Å, (1.7 μm, 2.1 mm × 150 mm) (Waters, Milford, CT, USA). Analyses were performed according to previous methods [13].

Data was processed using Masslynx 2.0 (Waters, Milford, CT, USA). Single components were identified by a comparison of an experimental deprotonated molecular (precursor) ion mass, mass fragmentation spectra, UV absorption spectra, the retention time to standards, and the literature data (articles and metabolite databases).

UV peaks were integrated into the range of 200–500 nm. For the statistical analysis of the chemical composition, the peaks of the UV chromatograms were integrated within the range of 200–500 nm. The area of integrated peaks was calculated as a percentage (%) of the combined area of all the peaks.

The area of these peaks was also used for the relative evaluation of component concentrations. The relative concentration was classified as trace (tr), + (low), ++ (average), and abundant (+++).

### 3.3. The Determination of Antimicrobial Activity

The propolis extracts dissolved in dimethyl sulfoxide (DMSO) were screened for antibacterial and antifungal activities by a micro-dilution broth method according to both the European Committee on Antimicrobial Susceptibility Testing (EUCAST) (www.eucast.org; accessed on 1 March 2021) [49] using the Mueller–Hinton broth, or RPMI with MOPS for the growth of fungi as we described elsewhere [10]. The minimal inhibitory concentrations (MIC) of the tested extracts were evaluated for the wide panel of the reference microorganisms from the American Type Culture Collection (ATCC), including Gram-negative bacteria (*Escherichia coli* ATCC 25922, *Salmonella* Typhimurium ATCC14028, *Klebsiella pneumoniae* ATCC 13883, *Pseudomonas aeruginosa* ATCC 9027, and *Proteus mirabilis* ATCC 12453), Gram-positive bacteria (*Staphylococcus aureus* ATCC 25923, *Staphylococcus aureus* ATCC 6538, *Staphylococcus epidermidis* ATCC 12228, *Micrococcus luteus* ATCC 10240, *Enterococcus faecalis* ATCC 29212, and *Bacillus cereus* ATCC 10876) and fungi (*Candida albicans* ATCC 10231, *Candida parapsilosis* ATCC 22019, and *Candida glabrata* ATCC 90030). The sterile 96-well polystyrene microtitrate plates (Nunc, Denmark) were prepared by dispensing 100 µL of the appropriate dilution of the tested extracts in a broth medium, per well, by serial two-fold dilutions, in order to obtain the final concentrations of the tested extracts that ranged from 0.0195 to 10 mg/mL. The inoculums that were prepared with fresh microbial cultures in sterile 0.85% NaCl, to match the turbidity of the 0.5 McFarland standard, were added to the wells to obtain a final density of 1.5 × 10^6^ CFU/mL for bacteria and 5 × 10^4^ CFU/mL for yeasts (CFU: colony forming units). After incubation (35 °C for 24 h), the MICs were assessed visually for the lowest concentration of the extracts, showing the complete growth inhibition of the reference microbial strains. An appropriate DMSO control (at a final concentration of 10%), a positive control (containing the inoculum without the tested derivatives), and the negative control (containing the tested derivatives without the inoculum) were included on each microplate.

The MIC for *H. pylori* ATCC 43504 was determined using a two-fold microdilution method in the MH broth with 7% of lysed horse blood at extract concentrations ranging from 1000 to 1.95 mg/L of bacterial inoculum with a 3 McFarland standard. After incubation at 35 °C for 72 h under microaerophilic conditions (5% O_2_, 15% CO_2,_ and 80% N_2_) the growth of *H. pylori* was visualized with the addition of 10 µL of 0.04% resazurin. The MIC endpoint was recorded after 4 h of incubation as the lowest concentration of the extract that completely inhibits growth [50].

The minimal bactericidal concentrations (MBC) or the minimal fungicidal concentrations (MFC) were obtained by a culture of 5 mL from each well that showed growth inhibitions from the last positive one, and from the growth control onto the recommended agar plates. The plates were incubated at 35° for 24 h for all microorganisms apart from *H. pylori,* which were incubated for 72 h in microaerophilic conditions.

The MBC/MFC was defined as the lowest concentrations of the extract without the growth of microorganisms. The MBC/MIC ratios were calculated to determine the bactericidal or bacteriostatic effects of the assayed extracts. Vancomycin, clarithromycin, ciprofloxacin, and nystatin were used as the reference drugs appropriate for different group of microorganisms (Appendix A).

The experiments were repeated in triplicate. Representative data are presented.

### 3.4. Statistical Analyses

Statistical analyses were performed by Statistica 14.0.0.5 software (Tibco Sofware Inc., Palo Alto, CA, USA). The correlation between the composition and the activity was analyzed by a correlation matrix. The matrix was composed of a percentage of UV chromatograms (200–500 nm), the relative peak area, and the antimicrobial activity (MIC values). Substances of at least 1% of the relative area (in any sample) were used to construct the matrix. Analyses included the components vs. MICs (Model 1) and components vs. components (Model 2). During the analyses, *r*, *p*, and *N* values were calculated. The obtained matrices were also used for the principal component analysis (PCA) and the hierarchical fuzzy clustering analysis (dendrogram). The PCA and dendrogram matrix was composed of only substances without MIC values. The prepared matrix was attached in Appendix A. Input data of statistical analyses).

## 4. Conclusions

In summary, all tested propolis extracts exhibited strong antibacterial activity against *Helicobacter pylori,* as well as the Gram-positive species included in the research. The in vitro anti-*H*. *pylori* activity of poplar and aspen–poplar propolises may be regarded as an important indication for the use of propolis as an effective agent in the eradication of *H*. *pylori*. Additionally, the application of propolis in external infections, especially caused by *Staphylococcus aureus*, could be an effective treatment. The results indicate the possibility of applying extracts from the tested propolis to protect food against microbiological spoilage, as well as the biocontrol of their growth, and they are similar to those obtained by Pobiega and collegues [51]. We also established that propolises of the poplar origin, with strong presence of flavonoid aglycones, are better antimicrobial agents than those consisting mainly of free phenolic acids and their glycerides. Therefore, the presence of these components in propolis may be a clue to the possible lowering of activities. However, in-depth research is required to draw further conclusions.

## Figures and Tables

**Figure 1 pathogens-11-00191-f001:**
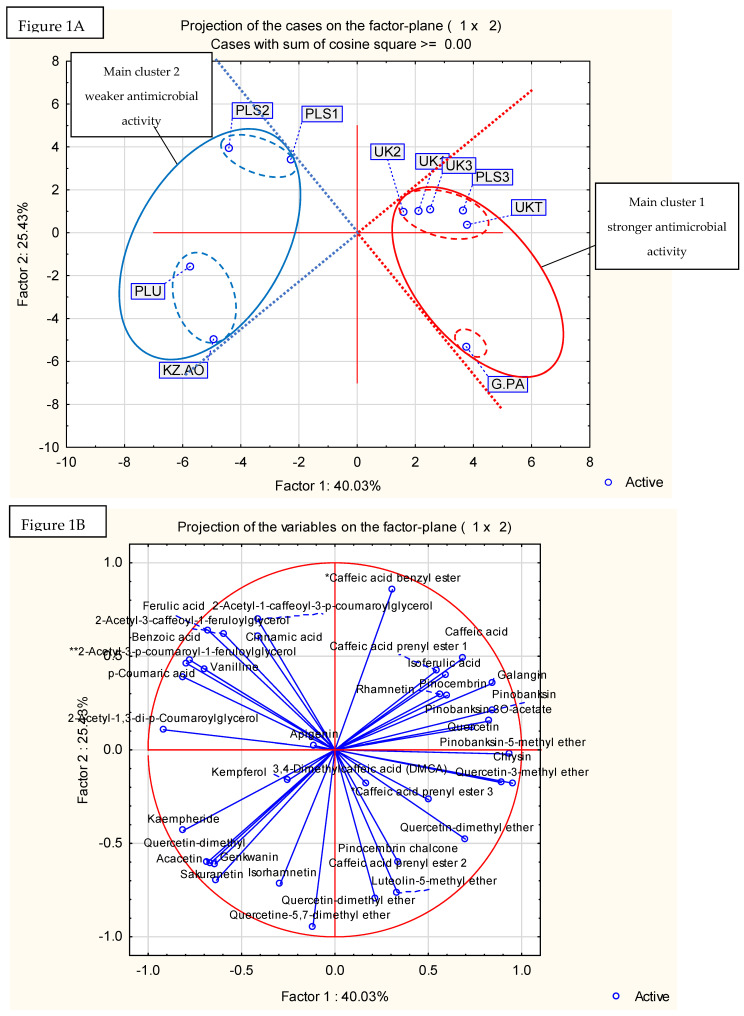
Principal component analysis. Projection of the cases on the factor plane (**A**) and projection of the variables on the factor plane (**B**).

**Figure 2 pathogens-11-00191-f002:**
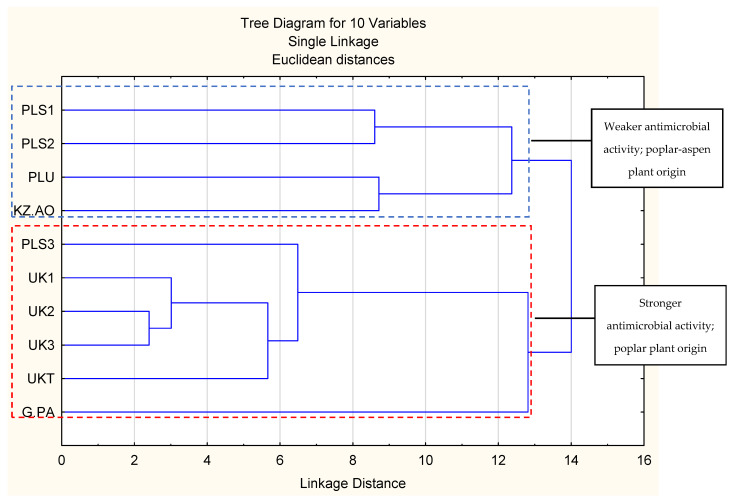
Hierarchical fuzzy clustering analysis presented as a dendrogram.

**Table 1 pathogens-11-00191-t001:** Relative concentrations of components of tested propolis samples.

Component	RT	UV λ Max (nm)	[M-H]-	PLS1	PLS2	PLS3	PLU	UK1	UK2	UK3	UT	KZ	GP
* Caffeoyl-glycerol	1.88	**324**, 298sh, 242	253	tr	tr	tr	tr	tr	tr	tr	tr	tr	tr
*p*-Hydroxy benzoic acid	2.13	**256**	137	tr	tr	tr	tr	tr	tr	tr	tr	tr	tr
Caffeic acid	2.17	**324**, 298sh, 242	179	+	+	+	tr	+	+	+	+	+	+
* *p*-Coumaroylglycerol	2.63	**310**, 300sh, 229	237	tr	tr	tr	tr	-	-	-	-	tr	-
*p*-Coumaric acid	3.25	**310**, 300sh, 229	163	++	+++	++	++	+	+	+	+	+	tr
Metoxybenzaldehyd	3.31	**276**	135	tr	tr	tr	-	-	-	-	-	-	-
Vanillin	3.42	**310**,280, 231	151	tr	+	tr	+	tr	tr	tr	-	tr	-
Ferulic acid	3.70	**324**, 298sh, 236	193	++	++	+	+	+	+	+	tr	+	tr
Isoferulic acid	4.11	**323**, 295sh, 221	193	+	tr	+	tr	+	tr	+	tr	tr	tr
Benzoic acid	5.97	281sh, 274sh, **236**	121	+	+	tr	+	+	+	+	tr	+	tr
* Ferulic acid derivate	6.82	**326**, 298sh, 236	389	-	-	-	tr	tr	tr	tr	tr	tr	-
Acetyl-*p*-coumraoylglycerol	7.40	**311**	279	tr	tr	tr	tr	tr	tr	tr	-	tr	-
3,4-Dimethylcaffeic acid (DMCA)	8.07	**322**, 294sh, 236	207	+	+	+	tr	tr	tr	tr	tr	tr	+
* Apigenin-O-glucoside	8.52	315sh, **265**	431	tr	tr	tr	-	tr	tr	tr	tr	tr	tr
Luteolin	11.30	**345**, 254	285	tr	tr	tr	-	tr	tr	tr	tr	tr	tr
Quercetin	11.43	368, 270sh, **256**	301	tr	tr	tr	tr	tr	tr	tr	+	tr	tr
Pinobanksin-5-methyl ether	12.32	322sh, **288**, 228	285	tr	tr	+	tr	+	+	+	+	tr	+
Cinnamic acid	12.94	**277**	147	tr	+	tr	tr	tr	tr	tr	-	-	-
Quercetin-3-methyl ether	13.68	355, 293sh, **255**	315	tr	tr	tr	tr	tr	tr	tr	tr	tr	+
* 1-Caffeoyl-3-*p*-coumaroyl glycerol	15.23	**315**, 298sh, 235	399	tr	tr	-	tr	-	-	-	-	tr	-
Naringenine	15.83	330sh, **290**, 230	271	tr	tr	tr	tr	tr	tr	tr	tr	tr	tr
Pinobanksin	16.30	332sh, **292**, 229	271	+	tr	+	tr	+	+	+	+	+	+
Apigenin	16.40	**338**, 290sh, 268	269	+	tr	+	+	+	+	+	tr	+	tr
Caffeoyl-feruloylglycerol	16.12	**326**, 298sh, 240	429	-	-	-	-	-	-	-	-	tr	-
Chrysin-5-met ether	16.80	314sh, **264,** 247sh	267	-	-	tr	tr	tr	tr	tr	tr	tr	tr
Kempferolhyl	17.48	366, 322sh, 295sh, 266	285	tr	+	+	+	+	+	+	+	+	tr
Isorhamnetin	19.11	371, 298sh, 268sh, **255**	315	tr	tr	tr	tr	tr	tr	tr	tr	+	tr
Quercetin-methyl ether	19.57	371, 298sh, 268sh, **255**	315	-	-	-	-	tr	tr	tr	-	tr	tr
Luteolin-5-methyl ether	20.15	350, 298sh, **266**, 232sh	299	tr	tr	+	tr	+	+	+	tr	+	+
** 1,3-Di-*p*-coumaroylglycerol	20.25	**310**, 300sh, 233	383	tr	tr	tr	tr	tr	tr	tr	tr	+	-
Quer-5,7-dimethyl ether	20.71	356, 296sh, 269sh, **255**	329	tr	tr	tr	tr	tr	tr	tr	tr	+	+
*p*-Coumaroyl-feruloylglycerol	21.37	**316**, 298sh, 233	413	tr	tr	-	-	-	-	-	tr	tr	-
Di-feruloiloglicerol	21.95	**323,** 298sh	443	tr	tr	-	-	tr	tr	tr	-	tr	-
2-Acetyl-1,3-di-caffeoylglycerol	22.57	**328**, 298sh, 244	457	tr	tr	-	tr	tr	tr	tr	tr	tr	-
β-styrylacrilic acid	23.82	**311**, 240sh	173	tr	tr	tr	tr	tr	tr	tr	tr	-	tr
Galangin-5-methyl ether	25.42	352, 300sh, **261**, 240sh	283	tr	tr	tr	-	tr	tr	tr	tr	tr	tr
Pinobanksin-5-methyl-ether-3-O-acetate	25.60	**289**	327	tr	tr	tr	-	tr	tr	tr	tr	tr	tr
* Caffeic acid buteniccetin or isobutenic ester	24.73	**326**, 298sh, 248	233	-	-	-	-	-	-	-	-	tr	tr
Rhamnetin	25.92	354, 298sh, 268sh, **255**	315	tr	tr	tr	tr	tr	tr	tr	+	tr	-
Quercetin-dimethyl ether	27.22	356, 292sh, 268sh, **256**	329	tr	tr	tr	tr	-	-	-	tr	tr	+
2-Acetyl-1-caffeoyl-3-*p*-coumaroylglycerol	29.23	**316**, 299sh 235	441	+	+	tr	tr	tr	tr	tr	tr	tr	-
Quercetin-dimethyl ether	29.46	356, 268sh, **256**	329	tr	tr	tr	tr	tr	tr	tr	tr	tr	+
Caffeic acid butyl or isobutyl ester	30.20	**326**, 298sh, 242	235	-	-	-	-	-	-	-	-	tr	tr
2-Acetyl-3-caffeoyl-1-feruloylglycerol	30.28	**328**, 300sh, 244	471	+	tr	tr	tr	tr	tr	tr	-	tr	-
Quercetin-trimethyl ether	30.63	353, 266sh, **254**	343	-	-	-	-	-	-	-	tr	-	tr
Caffeic acid prenyl ester 1	31.92	**326**, 298sh, 246	247	+	tr	+	tr	tr	+	+	+	tr	tr
Chrysin	32.11	314sh, **268,**	253	+	+	+++	+	+++	+++	+++	+++	+	+++
Pinobanksin-7-methyl ether	33.62	**291**, 247sh	285	-	-	re	-	tr	tr	tr	-	-	tr
Pinocembrin	33.68	330sh, **290**,	255	++	++	++	++	+++	+++	+++	+++	++	+
Acacetin	34.08	**335**, 299sh, 268	283	tr	tr	tr	+	-	-	-	-	+++	-
Caffeic acid prenyl ester 2	34.10	**326**, 298sh, 246	247	-	-	-	-	tr	tr	tr	-	tr	+
Pinocembrin chalcone	34.14	**345**	255	-	-	-	-	-	-	-	-	-	+
* Caffeic acid prenyl ester 3	34.21	**326**, 298sh, 246	247	+	+	+	-	tr	tr	tr	+	tr	++
* Caffeic acid benzyl ester	34.62	**328**, 298sh, 244	269	++	+	++	tr	+	+	+	++	tr	tr
Caffeic acid prenyl ester 4	34.58	**326**, 298sh, 246	247	tr	tr	tr	tr	tr	tr	tr	-	tr	tr
Sakuranetin	35.16	**290**	285	tr	tr	tr	++	+	+	+	+	+++	+
Genkwanin	35.64	337, **267**, 242sh	283	tr	tr	tr	+	-	-	-	tr	+	tr
Galangin	36.07	360, 290sh, **266**,	269	++	+	+++	+	+++	+++	+++	+++	+	++
Kaempheride	37.31	366, 292sh, **266**,	299	+	+	tr	++	tr	tr	tr	tr	++	tr
2-Acetyl-1,3-di-*p*-coumaroylglycerol (lasiocarpin A)	37.55	360sh, **312**, 232	425	+	++	tr	+	tr	tr	tr	tr	++	-
Pinobanksin-3O-acetate	37.82	332sh, **294**, 238	313	+	+	++	+	+++	+++	+++	++	+	++
Quercetin-dimethyl	38.33	370, 268sh, 255	329	-	-	-	+	-	-	-	-	+	tr
** 2-Acetyl-3-*p*-coumaroyl-1-feruloylglycerol	38.96	**318**, 299sh 235	455	+	+	-	+	tr	tr	tr	tr	+	-
Metoxychrysin	39.27	340sh, 310sh, **266**	283	tr	-	+	-	+	+	+	+	tr	+
3-Acetyl-1,2-di-*p*-coumaroylglycerol	40.09	**312**, 300sh, 238	425	tr	+	-	tr	tr	tr	tr	-	tr	+
Caffeic acid phenethyl ester (CAPE)	40.15	**326**, 298sh, 264	283	+	tr	+	-	+	+	+	+	+	+
2-Acetyl-1,3-di-feruloylglycerol	40.37	**328**, 298sh, 243	485	+	+	tr	tr	tr	+	tr	-	tr	-
Dimethyl luteolin	41.05	348, **267**, 246sh	313	-	-	-	+	tr	tr	tr	tr	++	-
Caffeic acid pentyl or isopentyl ester	41.90	**326**, 298sh, 247	249	-	-	-	-	-	-	-	-	-	-
Flavonoid dimethyl ether	42.53	**343,** 271, 248sh	343	-	-	-	-	tr	tr	tr	-	+	-
*p*-Coumaric acid prenyl ester 1	45.42	**311**, 299sh, 245sh	231	-	-	tr	-	tr	tr	tr	tr	-	-
*p*-Coumaric acid prenyl ester 2	45.42	**311**, 299sh, 245sh	231	+	+	tr	+	tr	tr	tr	tr	-	tr
*p*-Coumaric acid benzyl ester	45.80	**312**, 298sh, 244sh	253	+	++	+	++	+	+	+	+	+	tr
* Ferulic or isoferulic acid benzyl ester	47.13	**326,** 298sh	283	+	+	+	+	+	+	+	+	+	+
Caffeic acid cinnamic ester	48.04	**326**, 300sh, 243	295	tr	tr	tr	tr	+	+	tr	tr	tr	tr
Pinobanksin-3-O-propanoate	48.58	329sh, **294**, 234	327	tr	tr	tr	-	tr	tr	tr	tr	tr	+
*p*-Coumaric acid phenethyl ester	49.21	**312**, 300sh,	267	+	+	tr	tr	-	tr	tr	-	-	tr
Pinostrobin chalcone	49.67	**345,** 309sh, 267	269	-	-	-	+	-	tr	tr	tr	tr	+
Tectochrysin (chrysin-7-methyl ether)	51.23	310sh, **268**	267	tr	-	+	-	+	+	tr	+	-	+
Pinostrobin (pinocembrin-5-methyl ether)	51.40	328sh, **290**,	269	tr	-	tr	+	tr	+	tr	tr	+	+
Pinobanksin-3-O-butanoate or isonutanoate	51.65	329sh, **294**, 234	341	tr	tr	+	tr	+	tr	+	+	+	+
Galangin-7-methyl ether	52.21	353, **268**	283	tr	-	tr	tr	+	+	tr	tr	+	+
Pinobanksin-3-O-pentanoate or isopentanoate	53.07	332sh, **293**, 242	355	tr	-	tr	-	tr	tr	tr	+	tr	+
Pinobanksin-3-O-pentenoate or isopentenoate	53.24	332sh, **293**, 242	353	-	-	tr	-	tr	tr	tr	tr	++	+
Pinobanksin-3-O-hexanoate or isohexanoate	54.09	**282**	369	-	-	-	-	tr	tr	tr	tr	tr	-
Metoxycinnamic acid cinnamyl ester	54.26	**280**	293	tr	tr	tr	tr	tr	tr	tr	tr	tr	tr

Table legend: “-“—component absent; tr—component presented as trace (ions); +—component present in relatively low concentrations; ++—component present in relatively average concentration; +++—component present in relatively high concentration; *—component tenatively identified; **—subtitution positioning of glycerol was tenatively identified; GP—Greece, Parga; KZ—Kazakhstan, Almastka Oblast; PLS1—Poland, Lower Silesia Region 1; PLS2—Poland, Lower Silesia Region 2; PLS3—Poland, Lower Silesia Region3; PLU—Poland, Lubelszczyzna Region; UK1—Ukraine, Khmelnitsky 1; UK2—Ukraine, Khmelnitsky 2; UK3—Ukraine, Khmelnitsky 3; UT—Ukraine, Tarnopol.

**Table 2 pathogens-11-00191-t002:** Evaluation of MIC and MBC/MFC of propolis extracts [mg/mL].

Propolis Extract/ Microorganism	PLS1		PLS2		PLS3		PLU		UK1		UK2		UK3		UKT		KZ		GP	
**Gram− bacteria**	**MIC**	**MBC**	**MIC**	**MBC**	**MIC**	**MBC**	**MIC**	**MIC**	**MBC**	**MIC**	**MBC**	**MBC**	**MIC**	**MBC**	**MIC**	**MBC**	**MIC**	**MBC**	**MIC**	**MBC**
** *H. pylori* **	**0.03**	**0.03**	**0.06**	**0.06**	**0.03**	**0.03**	**0.03**	**0.03**	**0.03**	**0.03**	**0.03**	**0.03**	**0.015**	**0.03**	**0.03**	**0.03**	**0.02**	**0.03**	**0.03**	**0.03**
*S. typhimurium*	1.25	>10	>10	>10	>10	>10	>10	5.00	>10	>10	>10	>10	>10	>10	>10	>10	>10	>10	>10	>10
*E. coli*	2.50	>10	>10	>10	>10	>10	>10	5.00	>10	>10	>10	>10	>10	>10	>10	>10	>10	>10	>10	>10
*P. mirabilis*	2.50	10.0	>10	>10	>10	>10	>10	5.00	10.0	>10	>10	>10	>10	>10	>10	>10	>10	>10	>10	>10
*K. pneumoniae*	5.00	>10	>10	>10	>10	>10	>10	5.00	>10	>10	>10	>10	>10	>10	>10	>10	>10	>10	>10	>10
*P. aeruginosa*	1.25	10.0	>10	>10	>10	>10	>10	1.25	10.0	>10	>10	>10	>10	>10	>10	>10	>10	>10	>10	>10
**Gram+ bacteria**	**MIC**	**MBC**	**MIC**	**MBC**	**MIC**	**MBC**	**MIC**	**MIC**	**MBC**	**MIC**	**MBC**	**MBC**	**MIC**	**MBC**	**MIC**	**MBC**	**MIC**	**MBC**	**MIC**	**MBC**
*S. aureus* S1	0.16	0.31	0.63	2.50	0.08	0.16	0.16	0.16	0.63	19.50	78.0	0.63	0.04	0.16	0.01	0.04	0.08	0.16	0.01	0.04
*S. aureus* S2	0.31	1.25	2.50	10.0	0.08	0.16	0.31	0.31	0.63	0.08	0.16	1.25	0.16	0.31	0.01	0.04	0.16	0.16	0.01	0.04
*S. epidermidis*	0.08	0.31	0.31	2.50	0.04	0.08	0.16	0.08	0.31	0.0195	0.08	0.31	0.04	0.16	0.01	0.04	0.08	0.16	0.01	0.04
*E. faecalis*	0.31	2.50	2.50	10.0	0.16	0.63	0.63	2.50	2.50	0.08	0.31	10.0	0.16	0.63	0.08	0.31	0.16	0.63	0.08	0.31
*M. luteus*	0.08	0.16	0.31	2.5	0.04	0.08	0.08	0.08	0.16	0.02	0.08	0.31	0.04	156	0.01	0.02	0.04	0.08	0.01	0.02
*B. cereus*	0.16	0.16	0.63	0.63	0.04	0.08	0.31	0.31	0.31	0.02	0.08	1.25	0.16	0.16	0.02	0.04	0.16	0.16	0.02	0.04
**Yeasts**	**MIC**	**MFC**	**MIC**	**MFC**	**MIC**	**MFC**	**MIC**	**MIC**	**MFC**	**MIC**	**MFC**	**MFC**	**MIC**	**MFC**	**MIC**	**MFC**	**MIC**	**MFC**	**MIC**	**MFC**
*C. albicans*	0.31	1.25	1.25	5.00	0.08	0.16	0.31	0.31	1.25	0.16	0.16	1.25	0.16	0.31	0.16	0.31	0.16	0.31	0.08	0.16
*C. parapsilosis*	0.31	2.50	1.25	10.0	0.04	0.31	0.31	0.31	5.00	0.08	0.31	5.00	0.08	0.63	0.16	0.63	0.08	0.63	0.02	0.16
*C. glabrata*	0.31	0.63	0.63	2.5	0.08	0.16	0.63	0.31	0.63	0.08	0.16	2.5	0.16	0.31	0.16	0.31	0.16	0.31	0.08	0.08

Table legend: MIC—minimal inhibitory concentration; MBC—minimal bactericidal concentration; MFC—minimal fungicidal concentration; GP—Greece, Parga; KZ—Kazahstan, Almastka Oblast; PLS1—Poland, Lower Silesia Region 1; PLS2—Poland, Lower Silesia Region 2; PLS3—Poland, Lower Silesia Region 3; PLU—Poland, Lubelszczyzna Region; UK1—Ukraine, Khmelnitsky 1; UK2—Ukraine, Khmelnitsky 2; UK3—Ukraine, Khmelnitsky 3; UT—Ukraine, Tarnopol. *S. aureus* S1—*S. aureus* ATCC 25923; *S. aureus* S2—ATCC 29213.

## Data Availability

Research data is available from authors.

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
