# Peer review of "The Antimicrobial Properties of Poplar and Aspen–Poplar Propolises and Their Active Components against Selected Microorganisms, including Helicobacter pylori"

_pathogens, 2022, doi:10.3390/pathogens11020191_

Round 1

Reviewer 1 Report

This is a study to show antibacterial activity of honey to various kinds of bacteria and fungi. Authors examined a lot of chemicals in the honey and found antibacterial activities. However, the composition of the paper is not written in a usual manner, for example, Methods are written after the results, and the results and Discussion are written simultaneously. So readers cannot easily understand the contents of the paper. In addition the paper aims to find antibacterial properties of honey extracts to Helicobacter pylori. However, there are very few descriptions about H. pylori instead of the descriptions about S, aureus, etc. Authors should focus on H. pylori-related matters and shorten the manuscript. Furthermore, there are a lot of spelling and grammar mistakes. Authors should ask somebody who use English naturally to improve and check again and again before submitting the manuscript.

Author Response

Reviewer 1

This is a study to show the antibacterial activity of honey to various kinds of bacteria and fungi. The authors examined a lot of chemicals in the honey and found antibacterial activities. However, the composition of the paper is not written in a usual manner, for example, Methods are written after the results, and the results and Discussion are written simultaneously. So readers cannot easily understand the contents of the paper. In addition, the paper aims to find antibacterial properties of honey extracts to Helicobacter pylori. However, there are very few descriptions about H. pylori instead of descriptions about S, aureus, etc. Authors should focus on H. pylori-related matters and shorten the manuscript. Furthermore, there are a lot of spelling and grammar mistakes. Authors should ask somebody who uses English naturally to improve and check again and again before submitting the manuscript.

A: We thank you for the review and valuable issues. The manuscript was improved according to the suggestions. However, some things should be explained. All changes were pointed in yellow.

This is a study to show the antibacterial activity of honey to various kinds of bacteria and fungi. The authors examined a lot of chemicals in the honey and found antibacterial activities. However, the composition of the paper is not written in a usual manner, for example, Methods are written after the results, and the results and Discussion are written simultaneously.

A: We are sorry, but the focus of this manuscript is propolis, not honey. These are two different bee products. We prepared the manuscript according to MDPI instructions. In the basic template, the methodology is after the discussion section. Moreover, MDPI politics does not require separate results and discussion sections. Therefore, as the manuscript is long we decided to connect both sections in one.

However, there are very few descriptions about H. pylori instead of descriptions about S, aureus, etc. Authors should focus on H. pylori-related matters and shorten the manuscript.

A: We wanted to compare the activity of the tested extracts against H. pylori to the wide panel of non-fastidious reference strains.

Or We wanted to show the anti-H. pylori activity of tested extracts in the context of the wide panel of non-fastidious reference strains.

Reviewer 2 Report

The manuscript "Antimicrobial properties and active components of poplar and aspen-poplar propolises against selected microorganisms including Helicobacter pylori" presents an interesting comparison of the biological activity of various propolis. The research concerns Helicobacter pylori, which is in line with the subject special issue Pathogens journals.

Detailed comments:

Introduction - it is worth adding information about propolis poplar and aspen-poplar (eg. Doi: 10.3390 / molecules24162965).

line 122 - explain what the abbreviation tr means.

line 294 - in the journal the unit is mg / mL, not mg * mL ^ -1

Table 2 - the font should be reduced so that the values ​​are in one line and undivided (now it is not readable).

The description of the results is very good, interesting statistical tests were used to explain the dependencies between the results.

It is a pity that the authors, when they determined the MIC and MBC, did not carry out further microbiological analyzes, such as determining the time-kill of the best extracts, which I propose to investigate in subsequent studies. 

Author Response

Reviewer 2

The manuscript "Antimicrobial properties and active components of poplar and aspen-poplar propolises against selected microorganisms including Helicobacter pylori" presents an interesting comparison of the biological activity of various propolis. The research concerns Helicobacter pylori, which is in line with the subject special issue Pathogens journals.

A: We thank you for the review and valuable issues. The manuscript was improved according to the suggestions. All changes were pointed in yellow.

Detailed comments:

Introduction - it is worth adding information about propolis poplar and aspen-poplar (eg. Doi: 10.3390 / molecules24162965).

line 122 - explain what the abbreviation tr means.

A: Corrected.

line 294 - in the journal the unit is mg / mL, not mg * mL ^ -1

A: Corrected.

Table 2 - the font should be reduced so that the values ​​are in one line and undivided (now it is not readable).

The description of the results is very good, interesting statistical tests were used to explain the dependencies between the results.

A: Corrected.

It is a pity that the authors, when they determined the MIC and MBC, did not carry out further microbiological analyzes, such as determining the time-kill of the best extracts, which I propose to investigate in subsequent studies. 

A: Thank you for the suggestion. We are going to extend our research in further studies.

Reviewer 3 Report

The manuscript entitled “Antimicrobial Properties and Active Components of Poplar and Aspen-Poplar Propolises against Selected Microorganisms Including Helicobacter pylori” have described compositions and antimicrobial properties of propolis, especially against H. pylori. The aim of this study is important and topical, especially in the context of increasing drug resistance of microorganisms.

The study was written carefully and well in terms of language. The proposed statistical methods were appropriately selected according to the experiment and allow for the proper analysis of the research results. Authors should correct manuscript according to the suggestion.

Minor issues

Line 468 - 476: MIC values was evaluated the same method as in case H. pylori? Authors should give more details, what was the positive and negative control in the MIC experiment for bacteria and yeast?

Line 477 microdilution range should be given

Line 481: resazurine concentration should be given. After the addition of resazurin results were read immediately?

Author Response

Reviewer 3

The manuscript entitled “Antimicrobial Properties and Active Components of Poplar and Aspen-Poplar Propolises against Selected Microorganisms Including Helicobacter pylori” have described compositions and antimicrobial properties of propolis, especially against H. pylori. The aim of this study is important and topical, especially in the context of increasing drug resistance of microorganisms.

The study was written carefully and well in terms of language. The proposed statistical methods were appropriately selected according to the experiment and allow for the proper analysis of the research results. Authors should correct manuscript according to the suggestion.

A: We thank for review and valuable issues. Manuscript was improved according to the suggestions. All changes were pointed in yellow.

Minor issues

Line 468 - 476: MIC values were evaluated the same method as in case H. pylori? The authors should give more details, what was the positive and negative control in the MIC experiment for bacteria and yeast?

A: Corrected and completed

Line 477 microdilution range should be given

A: Corrected and completed

Line 481: resazurine concentration should be given. After the addition of resazurin results were read immediately?

A: Corrected and completed

Round 2

Reviewer 1 Report

The manuscript was revised in some parts, but the main body was not much changed.

1-There are still a lot of spelling and grammatical errors.

2-The manuscript is too long. I think the paper should be divided into two manuscripts: for Helicobacter and for other bacteria.

3-Tables are not good, please further edit the tables and complete the footnotes.

4-Helicobacter pylori are not Gram-positive bacteria.

5-Authors should read the manuscript again and again, and repair the misspellings and grammatical mistakes before submission.

Author Response

The manuscript was revised in some parts, but the main body was not much changed.

1-There are still a lot of spelling and grammatical errors.

2-The manuscript is too long. I think the paper should be divided into two manuscripts: for Helicobacter and for other bacteria.

3-Tables are not good, please further edit the tables and complete the footnotes.

4-Helicobacter pylori are not Gram-positive bacteria.

5-Authors should read the manuscript again and again, and repair the misspellings and grammatical mistakes before submission.

The answers:

AD 1. The manuscript has been reviewed by several people.  A lot of spelling and grammatical errors have been corrected and eliminated.

AD2. After discussion, the authors decided that from the beginning manuscript is not too long. All authors’ opinion is that the paper should not be divided into two manuscripts even though IF would be double.

AD 3. The tables are corrected.

AD 4. Of course, Helicobacter pylori is a Gram-negative bacterium. None of the authors claim otherwise.

AD 5. The authors read the manuscript again and again and corrected all mistakes before submission.
